# Direct Post-Training Preference Alignment for Multi-Agent Motion Generation Models Using Implicit Feedback from Pre-training Demonstrations

**Ran Tian**[1,2] , **Kratarth Goel**[2]
[1]UC Berkeley     [2]Waymo

## Abstract

Recent advancements in Large Language Models (LLMs) have revolutionized motion generation models in embodied applications such as autonomous driving and robotic manipulation. While LLM-type auto-regressive motion generation models benefit from training scalability, there remains a discrepancy between their token prediction objectives and human preferences. As a result, models pre-trained solely with token-prediction objectives often generate behaviors that deviate from what humans would prefer, making post-training preference alignment crucial for producing human-preferred motions. Unfortunately, post-training alignment requires extensive preference rankings of motions generated by the pre-trained model, which are costly and time-consuming to annotate—especially in multi-agent motion generation settings. Recently, there has been growing interest in leveraging expert demonstrations previously used during pre-training to scalably generate preference data for post-training alignment. However, these methods often adopt an adversarial assumption, treating all pre-trained model-generated samples as unpreferred examples and relying solely on pre-training expert demonstrations to construct preferred examples. This adversarial approach overlooks the valuable signal provided by preference rankings among the model's own generations, ultimately reducing alignment effectiveness and potentially leading to misaligned behaviors. In this work, instead of treating all generated samples as equally bad, we propose a principled approach that leverages implicit preferences encoded in pre-training expert demonstrations to construct preference rankings *among* the pre-trained model's generations, offering more nuanced preference alignment guidance with zero human cost. We apply our approach to large-scale traffic simulation (more than 100 agents) and demonstrate its effectiveness in improving the realism of pre-trained model's generated behaviors, making a lightweight 1M motion generation model comparable to state-of-the-art large imitation-based models by relying solely on implicit feedback from pre-training demonstrations, without requiring additional post-training human preference annotations or incurring high computational costs. Furthermore, we provide an in-depth analysis of preference data scaling laws and their effects on over-optimization, offering valuable insights for future studies.

# 1 Introduction

The recent advances in Large Language Models (LLMs) (Achiam et al., 2023) have significantly impacted the design of motion generation models for embodied tasks such as autonomous driving (Seff et al., 2023; Philion et al., 2024), robot manipulation (Brohan et al., 2023), and humanoid locomotion (Radosavovic et al., 2024). Formulating motion generation as a next-token prediction task not only provides a unified framework for modeling sequential decision-making tasks but also provides opportunities for leveraging pre-trained LLMs for more cost-effective training and improved generalizability (Tian et al., 2024). However, despite the remarkable progress, robots relying on these large token-prediction models do not automatically become better at doing what *humans* prefer due

Figure 1: **Direct preference alignment from occupancy measure matching feedback for realistic traffic simulation.** DPA-OMF is a simple yet effective alignment-from-demonstration approach that aligns a pre-trained traffic simulation model with human preferences. It defines an implicit preference distance function that measures the alignment between a generated sample and an expert demonstration in the same scene context through occupancy measure matching. This distance is then used to rank the reference model's generated samples for each training scene contex, enabling large-scale automatic preference data generation to align the motion generation model. The gray dotted lines above the motion token prediction model indicate the reference model's motion token distributions at each prediction step, and the orange lines represent the probabilities after the alignment process. $\hat{a}_t$ denotes agents' action tokens sampled from the predicted distribution $\mathbb{P}(\mathbf{a}_t; c, \hat{\mathbf{a}}_{t-1})$ during inference time, $c$ denotes the scene context representation (more details in Section 3.1).

to the misalignment between the training objective and the underlying reward function that incentivizes expert demonstrations. This discrepancy underscores the challenge of ensuring that motion models trained with next-token prediction are effectively aligned with expert-preferred behaviors.

Preference-based alignment has emerged as a crucial component in the LLM post-training stage, aiming to reconcile the disparity between the next-token prediction objective and human preferences. Among various frameworks, direct alignment algorithms (e.g., direct preference optimization (Rafailov et al., 2024b)) are particularly appealing due to their training simplicity and computational efficiency. Specifically, these algorithms collect human preferences over pre-trained model generations and directly update the model to maximize the likelihood of preferred behaviors over unpreferred ones. However, in complex embodied settings, such as joint motion generation involving hundreds of agents, obtaining such preference data at scale can be very challenging. Human annotators must analyze intricate and nuanced motions, which is a time-consuming process, making the scalability of direct alignment methods difficult in these scenarios.

While soliciting rankings from experts provides explicit preference information, we argue that expert demonstrations used in the pre-training stage inherently encode implicit human preferences, which can be reused to align a pre-trained motion generation model in a cost-effective way, beyond their role in supervised pre-training. Recently, Alignment from Demonstrations (AFD) (Li et al., 2024; Sun & van der Schaar, 2024; Chen et al., 2024b) has emerged as a valuable technique for automatically generating preference data using pre-training expert demonstrations, allowing preference alignment to scale at a low cost. However, previous methods typically assume a worst-case scenario: treating all pre-trained model-generated samples as unpreferred and relying solely on pre-training expert demonstrations to construct preferred examples (Chen et al., 2024b; Sun & van der Schaar, 2024). This adversarial approach overlooks the valuable signal provided by preference rankings among the model's own generations, ultimately reducing alignment effectiveness and potentially leading to misaligned behaviors.

*Instead of treating all generated samples as equally bad, we propose leveraging the implicit preferences encoded in pre-training demonstrations to automatically construct preference rankings **among** the pre-trained model's generations, providing more nuanced guidance with zero human cost.*

Our approach draws inspiration from inverse reinforcement learning (Abbeel & Ng, 2004; Ho & Ermon, 2016), where alignment between a generated behavior and the expert demonstration is measured through occupancy measure matching. We propose Direct Preference Alignment from Occupancy Measure Matching Feedback (DPA-OMF), a principled approach using optimal transport to define an implicit preference distance function. This function measures the alignment between a generated sample and the expert demonstration through occupancy measure matching in a seman-

tically meaningful feature space, and is then used to rank the generated samples according to their alignment with expert demonstrations, producing more nuanced preference data at scale to align the motion generation model (Figure 1).

We apply our approach to large-scale traffic simulation (more than 100 agents) and demonstrate its effectiveness in improving the realism of pre-trained model's generated behaviors, making a lightweight 1M motion generation model comparable to state-of-the-art large imitation-based models by relying solely on implicit feedback from pre-training demonstrations, without requiring additional post-training human preference annotations or incurring high computational costs.

> **Contribution**. In this work, we consider the problem of efficient post-training alignment of a token-prediction model for multi-agent motion generation. We propose Direct Preference Alignment from Occupancy Measure Matching Feedback (DPA-OMF), a simple yet principled approach that leverages pre-training expert demonstrations to generate implicit preference feedback and significantly improves the pre-trained model's generation quality without additional post-training human preference annotation, reward learning, or complex reinforcement learning. To the best of our knowledge, this is the first work to demonstrate the benefits of preference alignment for large-scale multi-agent motion generations using implicit feedback from pre-training demonstrations. Additionally, we provide a detailed analysis of preference data scaling laws and their impact on preference over-optimization.

## 2   Related Works

**Scaling alignment using AI feedback.** Previous works have proposed leveraging synthetic AI feedback to scale preference data in LLM applications, either from a single model (Bai et al., 2022; Lee et al.; Mu et al.), an ensemble of teacher models (Tunstall et al., 2023), or through targeted context prompting, where the model is instructed to generate both good and bad outputs for comparison (Yang et al., 2023; Liu et al., 2024). Unfortunately, these frameworks are not directly applicable to embodied contexts, such as autonomous driving, due to the absence of high-quality, open-source, and input-unified foundation models. What makes input-unified foundation models especially challenging is that different embodied models often use incompatible input modalities or features, making it difficult to transfer feedback across models effectively.

**Alignment from expert demonstrations.** Alignment from Demonstrations (AFD) has emerged as a valuable technique for automatically generating preference data using expert demonstrations, enabling preference alignment to scale effectively. For example, Chen et al. (2024b) fine-tunes a pre-trained reference model in a self-play manner, where the optimized model maximizes the log-likelihood ratio between expert demonstrations and self-generated samples. However, this method treats all model-generated samples as unpreferred, overlooking the valuable information embedded in the preference rankings among those samples. Additionally, it suffers from bias introduced by the *heterogeneity* of the preference data: since the preference data are drawn from two different sources (human vs. optimized model), the discrimination objective may emphasize the differences between models rather than focusing on the key aspects that truly evaluate the quality of the generated behaviors. To address the heterogeneity issue, Sun & van der Schaar (2024) proposes using expert demonstrations to first fine-tune the reference model through supervised learning, creating an expert model. Then, samples generated by the fine-tuned expert model are treated as positive examples, while samples from the initial model are treated as negative examples to construct a preference dataset. While this approach helps mitigate the heterogeneity problem, it requires training an additional expert model, and there is no guarantee that all generations from the expert model will consistently be superior to those from the initial model. *Unlike previous works, our approach constructs preference rankings among the generations produced by the reference model, effectively addressing the heterogeneity issue while providing more nuanced guidance at a lower cost.*

**Motion alignment via divergence minimization.** Adversarial imitation learning (IL) aims to align the behavior of the learning agent with expert demonstrations by minimizing the Jensen-Shannon divergence between the agent's action occupancy measure and that of the expert (Ho & Ermon, 2016; Song et al., 2018). These methods jointly train a policy model and a discriminator: the policy model is trained to imitate expert demonstrations, while the discriminator is trained to separate the generated samples from the expert demonstrations and informs the policy model. Additionally, adversarial training is known for its instability and high computational cost (Kodali et al., 2017; Yang

et al., 2022), making it particularly problematic in post-training alignment, where stability and computational efficiency are crucial. *Similar to adversarial IL, we employ occupancy measure matching to assess the alignment between generated samples and expert demonstrations. However, our approach differs by using preference rankings over the generated samples to inform the alignment process, rather than relying on a signal learned from separating all generated samples from expert demonstrations. Our approach provides more informative guidance while also being significantly more computationally efficient and more suitable for post-training alignment.*

**Preference alignment for realistic traffic simulation.** Preference-based alignment has recently been used to enhance the realism of traffic generation (Cao et al., 2024; Wang et al., 2024b). However, these previous works rely on Reinforcement Learning with Human Feedback (RLHF) as the alignment approach and learn rewards from low-fidelity simulator data. In contrast, our work seeks to improve traffic generation models without the need for reinforcement learning or explicit reward learning. Instead, we utilize an implicit preference distance derived from real human demonstrations to guide the alignment, and demonstrate improvements in a much larger scale.

# 3 EFFICIENT POST-TRAINING PREFERENCE ALIGNMENT FOR MULTI-AGENT MOTION GENERATION

## 3.1 PRE-TRAINING OF MULTI-AGENT MOTION GENERATION MODEL

We assume access to a set of expert demonstrations, $\mathcal{D}_e$, where each example consists of a joint action sequence $\xi = \{\mathbf{a}_0, \ldots, \mathbf{a}_T\}$ and a scene context representation $c$. The joint action sequence represents the behaviors of $N$ agents over a generation horizon $T$, conditioned on the given scene context $c$. Each joint action at time step $t$ is a collection of action tokens for all $N$ agents, denoted as $\mathbf{a}_t = (a_t^1, \ldots, a_t^N)$. Following Seff et al. (2023), our learning objective is to train a generative model parameterized by $\theta$, which maximizes the likelihood of the ground truth joint action at time step $t$, conditioned on all previous joint actions and the initial scene context:

$$\max_{\theta} \mathbb{E}_{(\xi,c)\sim\mathcal{D}_e} \left[ \Pi_{t=0}^T \pi_\theta(\mathbf{a}_t | \mathbf{a}_{<t}; c) \right], \tag{1}$$

where $\pi_\theta(\mathbf{a}_t | \mathbf{a}_{<t}; c) := \mathbb{P}(\mathbf{a}_t | \mathbf{a}_{t-1}, \ldots, \mathbf{a}_0; c)$. We denote the model pre-trained with the above token prediction objective as the reference model $\pi_{ref}$. Note that the optimal solution to (1) corresponds to finding a policy $\pi$ that minimizes the forward Kullback-Leibler (KL) divergence from the expert policy: $\min_\theta \text{KL}(\pi_e || \pi_\theta)$. As a result, the learned policy typically exhibits mass-covering behavior (Wang et al., 2024a), making it prone to deviations from the expert's true policy. This misalignment necessitates post-training preference alignment.

## 3.2 IMPLICIT PREFERENCE FEEDBACK FROM PRE-TRAINING DEMONSTRATIONS

In this section, we introduce our approach to leveraging pre-traning expert demonstrations for scalable preference feedback generation. The key idea is to define an implicit preference distance function that measures the alignment between a generated sample and the expert demonstration given the same scene context. This distance is then used to rank the reference model's generated samples for each training scene context.

Specifically, we aim to find a preference distance function $d$ that approximately measures the preference-based notion of similarity over a triplet $(\xi_e, \xi_s^i, \xi_s^j)$, where $\xi_e$ represents a pre-training expert demonstration and $\xi_s^{i,j}$ are two rollouts sampled from the reference model given the scene context of $\xi_e$. We treat $\xi_e^i$ as an anchor; if the preference distance between $\xi_s^i$ and $\xi_e$ is smaller than that between $\xi_s^j$ and $\xi_e$, then $\xi_s^i$ is considered more aligned with the anchor than $\xi_s^j$ and is thus more preferred according to the expert preference encoded in $\xi_e$: $d(\xi_e, \xi_s^j) > d(\xi_e, \xi_s^j) \implies \xi_s^i \succ \xi_s^j$.

Fundamental work in IRL (Abbeel & Ng, 2004) advocates for using occupancy measure matching to assess the alignment between the policy induced by a recovered reward function and human preference. A policy whose occupancy measure closely matches that of a human is considered better aligned with the human's true preferences Building on this insight, we leverage the Optimal Transport (OT) method (Villani et al., 2009) as a principled approach to define an implicit preference distance function that measures the occupancy measure matching between a policy rollout and the expert demonstration. We then use this preference distance to rank rollouts from the reference

model to construct the preference dataset. OT has been successfully used to measure alignment between behaviors in prior single-agent reinforcement learning works (Xiao et al., 2019) with the key difference in our work being used in multi-agent settings to fine-tune a generative model without the need for reinforcement learning (see Q4 in Appendix A for more details).

**Occupancy measure**. Let $\mathbf{o} = \{o_t\}_{t=0}^T$ represent a sequence of scene observations obtained by rolling out a joint action sequence $\xi$ in the initial scene $c$ over $T$ time steps. The empirical occupancy measure associated with $(\xi, c)$ is a discrete probability measure defined as $\rho_{(\xi,c)} = \frac{1}{T} \sum_{t=0}^T \delta_{\Phi(o_t)}$, where $\delta_{\Phi(o_t)}$ is a Dirac distribution centered on $\Phi(o_t) = [\phi(o_t^1), \ldots, \phi(o_t^N)]$. Here, $\phi$ is a feature encoder that maps each agent's information into a feature vector. Intuitively, a policy rollout occupancy measure represents the distribution of features visited by the generation policy throughout the generation horizon.

**Implicit preference distance.** Optimal transport quantifies the distance between two occupancy measures by solving the optimal coupling $\mu^* \in \mathbb{R}^{T \times T}$ that transports a rollout occupancy measure, $\rho_{(\xi_s,c)}$, to the expert occupancy measure, $\rho_{(\xi_e,c)}$, with minimal cost. Instead of computing the scene-level optimal coupling, we compute the agent-level optimal coupling and then aggregate them for computing the scene-level alignment, assuming the generative model only generates the behaviors of pre-defined agents but not insert nor remove agents from the scene. Specifically, for each agent $i$, we compute optimal coupling between agent $i$'s empirical occupancy measure induced by the sampled rollout $\xi^i$ and agent $i$'s empirical occupancy measure from the demonstration $\xi_e^i$ by minimizing the Wasserstein distance between the two:

$$\mu^{i,*} = \underset{\mu \in \mathcal{M}(\rho_{(\xi_e^i,c)}, \rho_{(\xi_s^i,c)})}{\arg\min} \sum_{t=1}^T \sum_{t'=1}^T \bar{c}\big(\phi(o_{t,s}^i), \phi(o_{t',e}^i)\big) \mu_{t,t'}. \tag{2}$$

where $\mathcal{M}(\rho_{(\xi_s^i,c)}, \rho_{(\xi_e^i,c)}) = \{\mu \in \mathbb{R}^{T \times T} : \mu\mathbf{1} = \rho_{(\xi_s^i,c)}, \mu^T\mathbf{1} = \rho_{(\xi_e^i,c)}\}$ is the set of coupling matrices, $\bar{c} : \mathbb{R}^n \times \mathbb{R}^n \to \mathbb{R}$ is a cost function defined on the support of the measure (e.g., cosine distance), and $n$ is the dimension of the feature space. This gives rise to the following distance that measures the preference-based notion of similarity between a rollout and the expert demonstration:

$$d(\xi_s, \xi_e; c) = \sum_{i=1}^N \sum_{t=1}^T \sum_{t'=1}^T \bar{c}\big(\phi(o_{t,e}^i), \phi(o_{t',e}^i)\big) \mu_{t,t'}^{i,*}. \tag{3}$$

**Preference data curation.** With the reference model pre-trained using the next-token prediction objective (1), we construct a rollout set, $\mathcal{D}_{\pi_{\text{ref}}}$, consisting of generated motions sampled from the model. Each example in $\mathcal{D}_{\pi_{\text{ref}}}$ includes a pre-training scene context $c$ (drawn from $\mathcal{D}_{\pi_e}$), the corresponding expert demonstration, and a set of $K$ rollouts, $\{\xi^1, \ldots, \xi^K\}$, sampled from the reference model given $c$. We use the implicit preference distance function defined in (3) to rank the $K$ rollouts of each training example in $\mathcal{D}_{\pi_{\text{ref}}}$ and construct the preference dataset $\mathcal{D}_{\pi_{\text{pref}}}$. Each example in $\mathcal{D}_{\pi_{\text{pref}}}$ contains $N_{\text{pref}}$ pairwise comparisons, $(\xi^+ \succ \xi^-)^{1, \ldots, N_{\text{pref}}}$, along with the associated scene context $c$. We note that this preference dataset is constructed without any human annotations, relying solely on pre-training expert demonstrations.

### 3.3 Direct preference alignment via contrastive preference learning

Using the automatically constructed preference dataset $\mathcal{D}_{\pi_{\text{pref}}}$, we apply a multi-agent extension of the contrastive preference learning algorithm (Hejna et al., 2023) to directly fine-tune the reference motion generation model:

$$\max_{\theta} \mathbb{E}_{(\xi^+, \xi^-, c) \sim \mathcal{D}_{\text{pref}}} \left[ -\log \frac{\exp \sum_t \gamma^t \alpha \log \frac{\pi_\theta(\mathbf{a}_t^+ | \mathbf{a}_{<t}^+, c)}{\pi_{\text{ref}}(\mathbf{a}_t^+ | \mathbf{a}_{<t}^+, c)}}{\exp \sum_t \gamma^t \alpha \log \frac{\pi_\theta(\mathbf{a}_t^+ | \mathbf{a}_{<t}^+, c)}{\pi_{\text{ref}}(\mathbf{a}_t^+ | \mathbf{a}_{<t}^+, c)} + \exp \sum_t \gamma^t \alpha \log \frac{\pi_\theta(\mathbf{a}_t^- | \mathbf{a}_{<t}^-, c)}{\pi_{\text{ref}}(\mathbf{a}_t^- | \mathbf{a}_{<t}^-, c)}} \right]. \tag{4}$$

## 4 Experiment Design

**Realistic traffic scene generation.** We validate our approach in the large-scale realistic traffic scene generation challenge (WOSAC), where the model is tasked with generating eight seconds of realistic

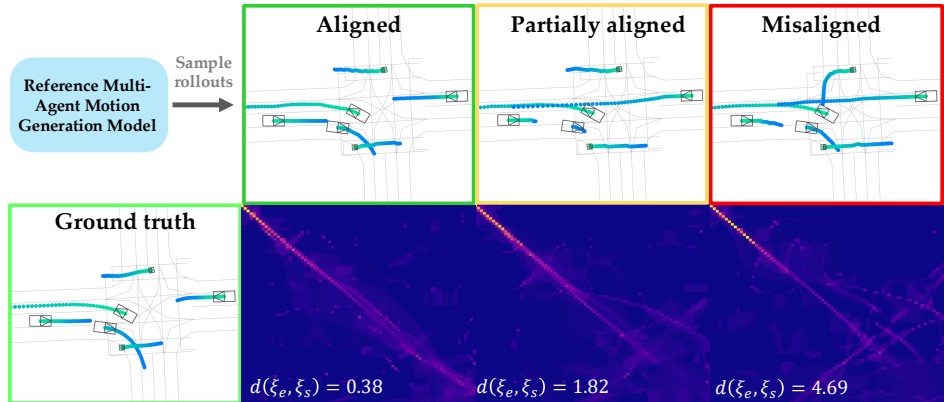

Figure 2: **Alignment visualization.** The heat map visualizes the optimal coupling between a generated traffic simulation and the ground truth scene evolution. More peaks along the diagonal indicate better alignment between the behaviors (i.e., a smaller preference distance). The traffic simulation with a small preference distance (left) shows behavior that is well-aligned with the ground truth, while the simulation with a larger distance (right) exhibits inconsistencies, such as the pedestrian's generated motion colliding with an oncoming vehicle.

interactions among multiple heterogeneous agents (up to 128) at 10 Hz, based on one second of past observations (Montali et al., 2024). The WOSAC challenge uses a realism metric to evaluate the quality of the traffic generation model. The realism of a generative model is defined as the empirical negative log likelihood of the ground truth scene evolution under the distribution induced by 32 scene rollouts sampled from that model (binned into discrete histograms).

**Token prediction model and preference dataset construction.** We use the MotionLM model (Seff et al., 2023) as our generation model (1M trainable parameters) (more details in Appendix C) and first train it using the next-token prediction objective on the Waymo Open Motion Dataset for 1.2M steps to obtain a reference model. For each training example, we sample 64 rollouts from the reference model and rank them using the preference distance function. The 16 closest rollouts are treated as preferred samples, while the 16 farthest are considered unpreferred, constructing 16 comparisons per example. Following previous work on learning rewards for autonomous driving (Sun et al., 2018; Chen et al., 2024a), we use features that capture the modeled agent's safety, comfort, and progress to encode the agent's state information at each time step when building the rollout occupancy measure, including: [collision status, distance to road boundary, minimum clearance to other road users, control effort, speed]. When solving the optimal transport plan (2), we use the L2 cost between features with the following weights: [10, 5, 2, 1, 1]. These features are also used to encode the agent's state in the realism metric. It is important to note that the preference distance is not the same as the realism metric. The preference distance measures the alignment between a rollout and the expert demonstration, whereas the realism metric assesses the likelihood of the expert demonstration given all the rollouts.

## 5  RESULTS

### 5.1  ON THE VALIDITY OF THE IMPLICIT PREFERENCE DISTANCE FUNCTION

In this section, we qualitatively and quantitatively investigate if the implicit preference distance reflects the behavior alignment (i.e., smaller distance means more aligned behavior).

**Qualitative example.** In Figure 2, we present a qualitative example illustrating how the coupling matrix reflects the behavior alignment between generated traffic simulations and expert demonstrations. Traffic simulations with smaller preference distances exhibit behavior more closely aligned with the expert demonstrations. Additional qualitative examples can be found in the AppendixD.

**Baseline**. To quantitatively evaluate our preference distance function in a controlled experiment, we conduct a post-selection analysis, where we select sampled traffic simulations from the reference model and analyze the relationship between the realism of these samples and their group-averaged distance to the expert demonstration. We use the Average Displacement Error (ADE) (L2 distance

between a sampled traffic simulation and the expert demonstration) as a baseline, which is a commonly used distance for evaluating trajectory generation performance in autonomous driving.

**Experiment setup**. We control the distance to the expert demonstration, and measure the realism of the selected sampled traffic simulations. Specifically, we first sample 128 traffic simulations from the pre-trained reference model, rank them using the candidate distance (ADE or preference distance), and then select traffic simulations as the final model output using a sliding window of size 32. For example, model variant 1 outputs the top 32 ranked traffic simulations, model variant 2 outputs simulations ranked from 2 to 33, and so on. We then measure the corresponding WOSAC realism of each model variant to study its relationship to each candidate distance metric.

**Results.** In Figure 3, we illustrate the relationship between the WOSAC realism of selected traffic simulations and their group-averaged distance from the expert demonstration. We observe that ADE is informative in reflecting behavior alignment only up to a certain point: initially, as ADE decreases, the realism improves. However, once the traffic simulations reach a reasonable level of realism, further reductions in ADE do not result in significant improvements in the realism. In contrast, the preference distance function correlates more strongly with the realism and demonstrates a better effective range when using the same reference model. This highlights its effectiveness in measuring the alignment between a generated traffic simulation and the expert demonstration.

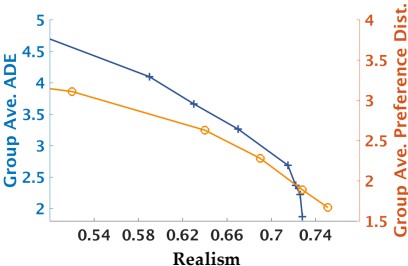

Figure 3: Relationship between the WOSAC realism of selected traffic simulations and their group-averaged distance from the expert demonstration.

## 5.2 ON THE VALUE OF PREFERENCE ALIGNMENT

In this section, we compare our approach with various post-training alignment methods to demonstrate its effectiveness. Our experiments focus on evaluating the impact of two factors: 1) different feedback signals, and 2) the alignment approach.

**Baseline.** We compare **DPA-OMF** against (1) **DPA-ADEF**, which constructs preference rankings using ADE as the distance function to rank traffic simulations sampled from the reference model, then fine-tunes the reference model using (4); (2) **SFT-bestOA**, which selects the top 32 ranked traffic simulations, measured by the preference distance, from the reference model and uses them as new labels for supervised fine-tuning, aiming to directly distill the preference into the reference model; (3) **SFT-bestADE**, similar to **SFT-bestOA** but use the ADE to pick the top ranked samples;(4) We also list the performance of SOTA models that are typically much larger with sophisticated designs. We use the same reference model for post-training alignment and fine-tune for 200k steps in all experiments.

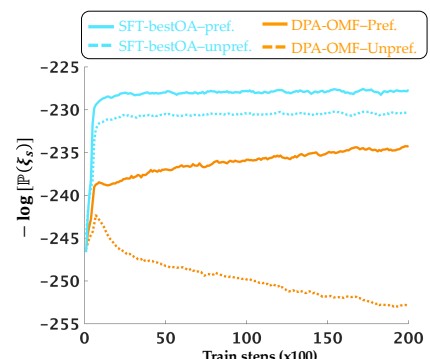

Figure 4: The log-likelihood of preferred/unpreferred rollouts from the reference model when using **DPA-OMF** versus **SFT-bestOA** for alignment.

**Results.** We present the quantitative evaluation of each method's realism and trajectory L2 error in Table 1. **DPA-OMF** significantly outperforms all baselines in terms of alignment between the generated traffic simulations and the expert demonstrations, while the baselines struggle to improve—and in some cases, even degrade (**DPA-ADEF** and **SFT-bestADE**)—the realism. Interestingly, **SFT-bestOA** does not seem to improve the realistic metric too much although it is using the same preferred traffic simulations as learning signals just like **DPA-OMF**. Figure 4 sheds more light on this. In Figure 4, we show the negative-loglikelihood of both preferred and unpreferred traffic simulations geenrated from the reference model when using **DPA-OMF** and **SFT-bestOA** to align the model. We can see that the likelihood of preferred traffic simulations is increasing and the likelihood of unpreferred traffic simulations is decreasing when using **DPA-OMF** to align the model, while the likelihood

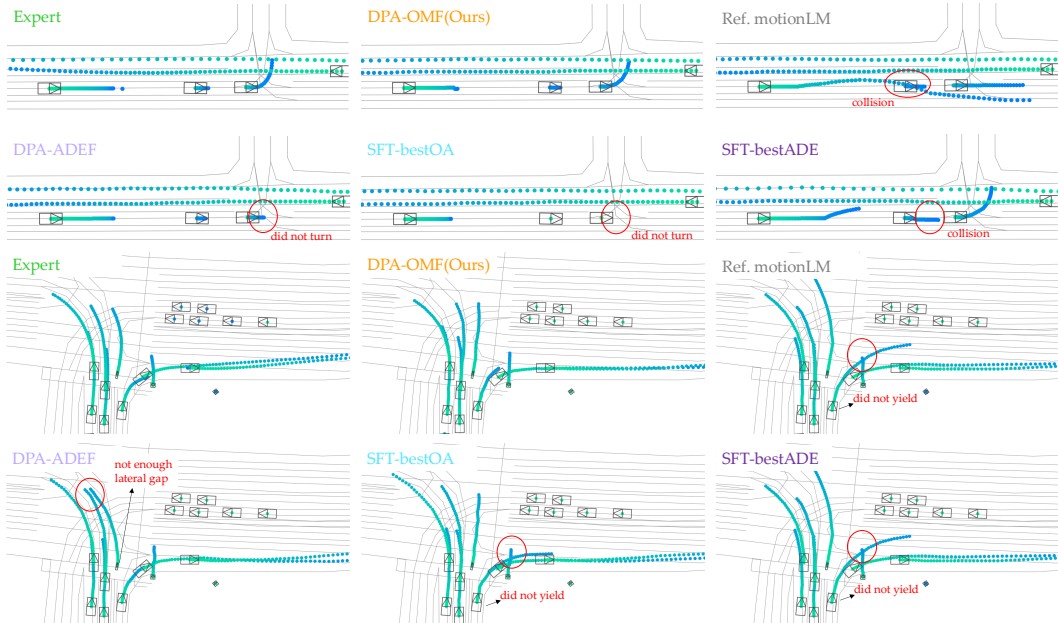

Figure 5: **Traffic simulation generation visualization.** Our approach produces traffic simulations that are more closely aligned with expert demonstrations, while baseline models generate simulations that are only partially aligned or misaligned (highlighted in red texts).

of unpreferred traffic simulations is increasing when using **SFT-bestOA** to align the model. This may be due to the fact that both the preferred and the unpreferred samples are from the same distribution. This demonstrates the importance of explicitly considering the negative signals when aligning the model. In Figure 5, we present a qualitative visualization of the generated traffic simulations from each approach. We sample 64 rollouts from each model and display the most likely generation. Our approach produces traffic simulations that are more closely aligned with expert demonstrations, whereas the baseline models generate simulations that are only partially aligned or misaligned. While our approach improves the reference model through cost-effective preference alignment, it still falls short a bit compared to some SOTA methods. We provide more discussions in $Q3$ of Appendix A.

| Method | Kinematic ↑ | Interactive↑ | Map compliance↑ | Composite realism↑ | minADE↓ |
|---|---|---|---|---|---|
| **MotionLM** (1M reference model) | **0.417** | 0.778 | 0.815 | 0.721 | 1.398 |
| **DPA-ADEF** | 0.393 | 0.780 | 0.812 | 0.714 | **1.379** |
| **SFT-bestADE** | 0.406 | 0.773 | 0.816 | 0.715 | 1.392 |
| **SFT-bestOA** | 0.410 | 0.781 | 0.826 | 0.723 | 1.428 |
| **DPA-OMF** (ours) | 0.415 | **0.786** | **0.867** | **0.739** | 1.413 |
| BehaviorGPT(Zhou et al., 2024) (3M) | 0.433 | 0.799 | 0.859 | 0.747 | 1.415 |
| Trajeglish(Philion et al., 2024) (35M) | 0.415 | 0.786 | 0.867 | 0.721 | 1.544 |
| SMART(Wu et al., 2024) (102M) | 0.479 | 0.806 | 0.864 | 0.761 | 1.372 |

Table 1: Realism of different methods. Our approach improves the realism of the reference model (shaded in gray) without requiring additional human-cost, reward learning or reinforcement learning.

**On the importance of features.** Despite the OT-based preference score's effectiveness in measuring the alignment between a generated traffic simulation and the expert demonstration, the relationship between the OT-based preference score and human preference is not strictly monotonic. This relationship heavily depends on the features used to compute the rollout occupancy measure. We consider a set of features commonly used in IRL research to compute the preference distance (described in Section 4). In Table 2, we present an ablation study analyzing the effect of each feature on the effectiveness of the approach. Although the best performance is achieved when all features are active, it is interesting to note that using only the collision status when computing the preference distance for model alignment still leads to improvements. We hypothesize that this is because the reference model already performs reasonably well in generating behaviors but lacks awareness of collisions. However, when using only the progress or comfort feature to compute the preference

distance, both the realism (due to an increased collision rate) and ADE regress. This highlights the importance of using a comprehensive set of features to accurately characterize driving behaviors. Further discussions on the limitations and potential solutions can be found in $Q1$ of Appendix A.

| Features | Kinematic | Interactive | Map compliance | Composite realism | minADE |
|---|---|---|---|---|---|
| **Collision only** | 0.389 | **0.788** | 0.833 | 0.724 | 1.527 |
| **Progress only** | **0.421** | 0.760 | 0.812 | 0.710 | 1.483 |
| **Comfort only** | 0.411 | 0.744 | 0.820 | 0.705 | 1.581 |
| **Full** | 0.415 | **0.786** | **0.867** | **0.739** | 1.413 |

Table 2: The effect of each feature on the effectiveness of our approach.

## 5.3 COMPARISON WITH ADVERSARIAL PREFERENCE ALIGNMENT FROM DEMONSTRATIONS

We further compare our method with the standard AFD approach, which treats all samples from the reference model as negative samples. For each training sample, we construct 16 rankings by sampling 16 generated traffic simulations from the reference model (i.e., both our method and AFD utilize the same amount of preference data). We measure the WOSAC realism of the fine-tuned model, the model's ability to assign higher likelihood to preferred traffic simulations ranked by our preference distance (measured as classification accuracy), and the minADE. As shown in Table 3, our approach significantly outperforms the adversarial AFD in all metrics, demonstrating the effectiveness of our method.

| Features | Classification accuracy ↑ | Composite realism ↑ | minADE ↓ |
|---|---|---|---|
| **Ours** | **0.84** | **0.739** | **1.413** |
| **Adversarial AFD** | 0.52 | 0.720 | 1.539 |

Table 3: The comparison between DPA-OMF with adversarial AFD. Our approach significantly outperforms the adversarial AFD in all metrics.

To further analyze why adversarial preference alignment is less effective, we plot the negative log-likelihood of expert demonstrations, preferred and unpreferred traffic simulations (ranked by our preference distance) in Figure 6. The results reveal that the likelihood of expert demonstrations is consistently much higher than that of both preferred and unpreferred samples throughout the alignment process. This stems from the pre-training phase, where expert demonstrations are used to train the reference model. Moreover, during the preference alignment phase, the model primarily increases the likelihood of expert demonstrations while leaving the likelihood of preferred and unpreferred samples relatively unchanged. This indicates that the model is unable to capture nuanced differences between preferred and unpreferred samples, leading to suboptimal alignment performance (see more analysis in Appendix F).

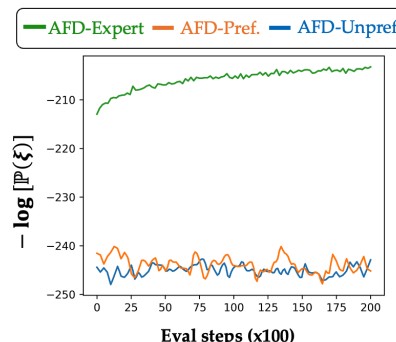

Figure 6: The log-likelihood of expert demos and preferred/unpreferred rollouts from the reference model when AFD for alignment.

## 5.4 PREFERENCE SCALING

One of the key advantages of **DPA-OMF** is its ability to leverage expert demonstrations to automatically construct preference rankings without requiring additional human annotations, making it highly scalable. In this section, we evaluate the performance of **DPA-OMF** as we scale the number of preference rankings. In Figure 7 (left), we show the relationship between the aligned model's performance and the average number of preference rankings per training example (e.g., a value of 4 indicates that the dataset used for alignment is four times larger than the original training set). Interestingly, we observe that a small amount of preference feedback not only fails to improve the

model but actually degrades it. However, as the number of preference rankings increases, the alignment objective begins to demonstrate its effectiveness. We hypothesize that this degradation is due to preference over-optimization, which we explore further in the next section.

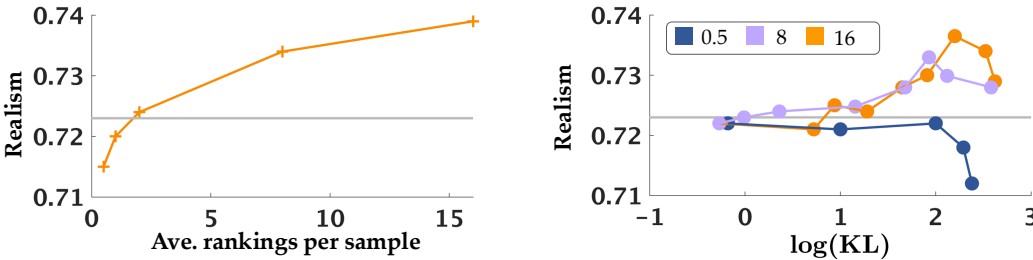

Figure 7: [**left - preference scaling**]: Performance of the alignment under different preference training data sizes. The gray line represents the performance of the reference model; [**right - preference over-optimization**]: The trade-off between the policy drift and the fine-tuning performance gain under various preference data sizes

.

## 5.5 PREFERENCE OVER EXPLOITATION

Preference over-optimization has been studied in the context of LLMs for both online methods (e.g., RLHF) and direct preference alignment methods (Tang et al., 2024; Rafailov et al., 2024a). In this section, we investigate this phenomenon in the context of multi-agent motion generation. While previous research has focused on the impact of preference over-optimization across different model sizes, our study examines its effects with varying data sizes.

Goodhart's law states that "when a measure becomes a target, it ceases to be a good measure" (Goodhart & Goodhart, 1984). In our case, this effect manifests when there is insufficient feedback, causing the model to over-optimize an incomplete preference signal. Following (Tang et al., 2024), we examine this effect by measuring the KL divergence between the reference model and the fine-tuned model at various preference data sizes during the alignment process. KL divergence quantifies how much the optimized policy deviates from the reference model's policy during preference learning, and can be interpreted as the optimization cost incurred by the alignment. In Figure 7 (right), with little preference data (e.g., 0.5 rankings per training example), the optimized policy drifts away from the reference but degrades performance. As we scale the preference data, the same policy drift budget results in better performance, though further increases in the KL budget eventually reduce performance. Nevertheless, this investigation shows that scaling preference data can mitigate the effects of preference over-optimization and highlights the importance of doing so in a cost-effective manner.

## 6 CONCLUSION

In this work, we consider the problem of efficient post-training alignment for multi-agent motion generation. We propose Direct Preference Alignment from Occupancy Measure Matching Feedback, a simple yet principled approach that leverages pre-training expert demonstrations to generate implicit preference feedback and improves the pre-trained model's generation quality without additional post-training human preference annotation, reward learning, or complex reinforcement learning. We presented the first investigation of direct preference alignment for multi-agent motion generation models using implicit preference feedback from pre-training demonstrations. We applied our approach to large-scale traffic simulation (more than 100 agents) and demonstrated its effectiveness in improving the realism of pre-trained model's generated behaviors, making a lightweight 1M motion generation model comparable to state-of-the-art large imitation-based models by relying solely on implicit feedback from pre-training demonstrations, without requiring additional post-training human preference annotations or incurring high computational costs. Additionally, we provided an in-depth analysis of preference data scaling laws and their effects on over-optimization, offering valuable insights for future investigations.

**Reproducibility Statement**. To enhance the reproducibility of our research, we have provided a detailed explanation of our motion generation model and its source in Appendix C. Additionally, we have thoroughly described the calculation of the preference distance in Section 3.2 and the process of constructing our preference dataset in Section 4.

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
