# OpenReview forum: "Direct Post-Training Preference Alignment for Multi-Agent Motion Generation Model Using Implicit Feedback from Pre-training Demonstrations"
_ICLR.cc/2025/Conference — ICLR 2025 Spotlight_

### Official Review · Reviewer_YM4s · 2024-10-18

**Soundness:** 3
**Presentation:** 1
**Contribution:** 2
**Rating:** 6
**Confidence:** 2

**Summary:**

The paper presents a novel approach inspired by inverse reinforcement learning, proposing **Direct Preference Alignment from Occupancy Measure Matching Feedback**. This method aims to align generated behaviors with expert demonstrations by matching occupancy measures in a semantically meaningful feature space. This method does not rely on additional human annotations or complex reinforcement learning but instead leverages the implicit preferences encoded in expert demonstrations. DPA-OMF ranks model-generated samples based on their alignment with expert behaviors using occupancy measure matching in a semantically meaningful feature space. The model is capable of handling up to **128 agents** using a **1M token-prediction model**.

#### Strengths:
1. **Scaling Experiments:** The paper includes comprehensive ablation studies, highlighting the model’s performance when scaling up the number of agents.
2. **Detailed Experimental Setup:** The authors provide thorough descriptions of the experimental setups, including parameters and conditions, contributing to the reproducibility and clarity of their results.

#### Weaknesses:
1. **Some miss proofs in the paper:** "These algorithms collect preference rankings from humans over model generations and directly update the model to maximize the likelihood of preferred behaviors over unpreferred ones." and " Human annotators must analyze intricate and nuanced motions, which is a time-consuming process, making the scalability of direct alignment methods difficult in these scenarios." need more proofs. I think there should be some citations or experiments to show.
2. **Overuse of Colors and Fonts:** The excessive use of different colors and fonts in the main text affects the readability and cohesiveness of the presentation. A more consistent design would improve the clarity of the paper.
3. **Visual Clarity of Images:** Some images in the paper are difficult to interpret due to potential resolution, contrast, or layout issues, which could hinder the reader’s ability to understand the visual data being presented.


I will refine this review based on the author's rebuttal and feedback from other reviewers. As the discussion progresses, further improvements or adjustments to the evaluation will be considered.

**Strengths:**

1. **Scaling Experiments:** The paper includes comprehensive ablation studies, which highlight the model’s performance when scaling up the number of agents.
2. **Detailed Experimental Setup:** The authors provide thorough descriptions of the experimental setups, including parameters and conditions, contributing to the reproducibility and clarity of their results.

**Weaknesses:**

1. **Some miss proofs in the paper:** "These algorithms collect preference rankings from humans over model generations and directly update the model to maximize the likelihood of preferred behaviors over unpreferred ones." and " Human annotators must analyze intricate and nuanced motions, which is a time-consuming process, making the scalability of direct alignment methods difficult in these scenarios." need more proofs. I think there should be some citations or experiments to show.
2. **Overuse of Colors and Fonts:** The excessive use of different colors and fonts in the main text affects the readability and cohesiveness of the presentation. A more consistent design would improve the clarity of the paper.
3. **Visual Clarity of Images:** Some images in the paper are difficult to interpret due to potential issues with resolution, contrast, or layout, which could hinder the reader’s ability to understand the visual data being presented.

**Questions:**

Current experiments have demonstrated the feasibility of the approach on a 1M-scale model. Will it still be effective on larger-scale models?

---

> ### Author Response · Authors · 2024-11-23
> **Responses to reviewer YM4s**
>
> We would like to thank the reviewer for taking the time to review our paper and provide valuable feedback. We are glad that the reviewer thinks our approach is novel and our results are supported by comprehensive ablation studies. We are excited to have the chance to address the reviewer’s questions and concerns. We have added additional clarifications in the paper and added new experiment results to address the reviewer’s questions. These edits will make the paper stronger.
>
> ***Q1.  Miss proof “These algorithms collect preference rankings from humans over model generations and directly update the model to maximize the likelihood of preferred behaviors over unpreferred ones”***
>
> A1. In this sentence, we aim to describe the high-level methodology of the direct preference alignment algorithm. To improve clarity and flow, we have revised the sentence to better connect with the preceding statement, ensuring that readers can more easily understand its context and meaning.
>
> ***Q2.  Miss proof “Human annotators must analyze intricate and nuanced motions, which is a time-consuming process, making the scalability of direct alignment methods difficult in these scenarios." need more proofs. I think there should be some citations or experiments to show”***
>
> A2. We would like to thank the reviewers for raising this question! We have addressed this in our general statement G3: we conducted a human subject study to show the substantial human cost associated with building preference data at scale.
>
> ***Q3.  Overuse of Colors and Fonts”***
>
> A3. We would like to thank the reviewers for the helpful comment. We have removed some of the colored words in the paper to improve the readability.
>
> ***Q4.  Visual Clarity of Images***
>
> A4. We have improved the resolution of some plots and added additional explanations in the figure caption to improve the visual understanding.

---

> > ### Comment · Reviewer_YM4s · 2024-11-24
> >
> > The authors have addressed most of my concerns during the rebuttal and I am happy to raise my score.

---

> > > ### Author Response · Authors · 2024-11-26
> > >
> > > Dear reviewer YM4s,
> > >
> > > We are glad that our responses could address your questions, and we appreciate your reconsideration on the score of our paper!
> > > Thank you again for taking the time to review our paper and providing the insightful comments!
> > >
> > > Best regards,
> > >
> > > Authors

---

### Official Review · Reviewer_3Bbi · 2024-11-03

**Soundness:** 2
**Presentation:** 4
**Contribution:** 2
**Rating:** 8
**Confidence:** 4

**Summary:**

This paper introduces a novel alignment from demonstration (AFD) strategy for multi-agent motion generation in the autonomous driving setting. Compared to direct annotation of preferences by humans, AFD scales better for the multi-agent setting. However prior AFD methods assume all base model's (or fine-tuned version of it) motion samples to be non-optimal while all demonstrations to be optimal. These alignment strategies are inefficient compared to the proposed method, which also compares the relative quality of generated samples among themselves. The paper shows improved alignment after their proposed AFD measured in terms of collision / progress / comfort features in the autonomous driving motion prediction tasks.

**Strengths:**

The paper is very well motivated and presented. The idea is novel and results are well explained by their visualizations. The proposed optimal transport based distance metric is compared to L2-distance baseline. Insights such as why their method works better than supervised fine-tuning as training continues, preference scaling, preference vs exploitations are also investigated.

**Weaknesses:**

1. There is a lack of discussion of the assumptions or limitations of the proposed method. For example, one assumption is that the OT-based distance between demonstration and generated samples captures the preferences in a monotonic fashion. Is this always the case in self-driving setting? Another assumption is that asking humans to provide dense trajectory demonstration for multiagent interactions is easier to just rank them (even though there are might be many more pair-wise rankings). Do you have any statistics or references that show the prior scale better than the latter annotation scheme?

2. The biggest concern is that there is limited if not no comparison with methods this paper is set out to improve: prior AFD methods that assume all base model's (or fine-tuned version of it) motion samples to be non-optimal. While the paper does compare their OT-distance metric works better than L2 distance as well as AFD works better than SFT, their main motivation of improving prior AFD methods is not validated.

3. While they show OT distance works better than L2 distance and AFD better than SFT, the improvement in table 1 results is quite incremental, limiting the contribution of the work.

**Questions:**

1. Can you compare your method to prior AFD methods and show both quantitatively and qualitatively (in figures) why comparing model generations among themselves help? Can you show some examples of the bias introduced by the heterogeneity of the preference data?
2. What are limitations of AFD? How many human direct annotations do you need vs how many demonstrations of how many cars do you need? How does alignment improves in terms of the labels provided in both cases? (scaling concern in multiagent setting)
3. If demonstrations are multi-modal, will your method of comparing sampled based of OT-distance metric introduce conflicting gradients and leading to mode collapse?
4. How does your OT-distance metric factor in collision / progress / comfort features?
5. Do you have qualitative figures that help readers understand why L2 distance drop at the end in Fig 3? Why does L2 distance metric will lead to missed turn in Fig5?

---

> ### Author Response · Authors · 2024-11-23
> **Responses to reviewer 3Bbi [1/2]**
>
> We would like to thank the reviewer for taking the time to review our paper and provide valuable feedback! We are glad that the reviewer thinks our paper is well-motivated and is novel. We are excited to have the chance to address the reviewer’s questions and concerns. In response, we have conducted additional experiments to address the reviewer’s comments on baselines and other aspects of the paper. These updates have been incorporated into the revised manuscript and, we believe, will further strengthen the paper.
>
> ***Q1.1  Discussion of the assumptions or limitations of the proposed method.***
>
> ***Q1.1  One assumption is that the OT-based distance between demonstration and generated samples captures the preferences in a monotonic fashion. Is this always the case in self-driving settings?***
>
> A1. We would like to thank the reviewer for this question! The OT distance quantifies the divergence between the occupancy measures of demonstrations and model generations. It has recently been used in the inverse reinforcement learning community to assess whether policy rollouts align with demonstrations (i.e., are more preferred) and has been shown to correlate well with human preferences through controlled experiments [1].
>
> However, we note that the relationship between OT distance and human preference is not strictly monotonic. This relationship heavily depends on the features used to compute the feature occupancy measure. For instance, if we only use safety features in the occupancy measure to rank reference model generations for alignment, we may observe that the model generates more safe motions, but the model may eventually become overly conservative, deviating from the true preferences of human drivers, as shown in Table 2. Further ablations on the impact of features can be found in Section 5.2.
>
> In this work, we leverage manually designed features that are well-validated and widely used in the autonomous driving industry. While these features allow for controlled experiments and reliable evaluation, they also limit the expressiveness of the features. We discuss this limitation in the motivating question Q1 in Appendix A  of the updated main manuscript and have updated the main text (Section 5.2) to explicitly highlight the advantages, implications, and limitations of using manually designed features, as well as potential solutions for future work.
>
> [1] Tian, Ran, et al. "What Matters to You? Towards Visual Representation Alignment for Robot Learning." ICLR, 2024.
>
> ***Q1.2 Another assumption is that asking humans to provide dense trajectory demonstration for multiagent interactions is easier to just rank them (even though there might be many more pair-wise rankings). Do you have any statistics or references that show the prior scale better than the latter annotation scheme?***
>
> A1.2 We appreciate the reviewer’s insightful comment and apologize for any confusion caused by our assumption about human demonstrations. To clarify, we are not claiming that asking humans to provide demonstrations is inherently easier than providing rankings. Instead, our intention is to highlight that scaling preference rankings in multi-agent interaction settings poses significant challenges.
>
> In our general statement G3, we conducted a human subject study to show the human cost required to manually label the preference data used in our experiments. Our motivation is to leverage existing demonstrations to construct preference rankings at scale, extending their traditional role in the pre-training phase. This approach allows us to bypass the limitations associated with scaling human-provided rankings in complex multi-agent scenarios.
> To address this point and avoid misunderstandings, we have updated the introduction to better reflect our assumptions and clarify this distinction.
>
> ***Q2.  Can you compare your method to prior AFD methods and show both quantitatively and qualitatively (in figures) why comparing model generations among themselves help? Can you show some examples of the bias introduced by the heterogeneity of the preference data?***
>
> A2. We appreciate the reviewer’s valuable suggestions and agree that a comparison with the suggested baseline is important. We have addressed these two questions with additional experiments results in the general statements G1 and G2.

---

> > ### Author Response · Authors · 2024-11-23
> > **Responses to reviewer 3Bbi [2/2]**
> >
> > ***Q3.  What are limitations of AFD? How many human direct annotations do you need vs how many demonstrations of how many cars do you need? How does alignment improves in terms of the labels provided in both cases? (scaling concern in multiagent setting)?***
> >
> > A3. In general, both pre-training and post-training preference alignment benefit from an increase in demonstration data, provided that the model capacity is properly scaled. In our experiment, we used all the Waymo open motion dataset for pre-training. Since our work focuses on the post-training preference alignment, we only demonstrated the effects of scaling preference data.
> >
> > While our work does not directly investigate sample efficiency of pre-training and scaling of expert demonstrations, we recognize their potential impact on alignment and are excited to explore these implications in future work, particularly the relationship between pre-training sample efficiency and post-training preference alignment.
> >
> >
> > ***Q4.  If demonstrations are multi-modal, will your method of comparing sampled based of OT-distance metric introduce conflicting gradients and leading to mode collapse?***
> >
> > A4. Thank you for this insightful question! As noted in LLM research, preference alignment methods (e.g., RLHF or contrastive preference update) inherently could lead to mode collapse and reduce diversity [1]. In such cases, current preference alignment algorithms may disregard minority preferences, leading to an overemphasis on majority preferences and a potential loss in output diversity.
> >
> > In our experiments, we observed that the fine-tuned model generates significantly fewer unsafe modes, reflecting the alignment’s effectiveness in suppressing undesirable behaviors. To assess the impact on mode diversity, we calculated the L2 distance between each pair of trajectory modes among 32 generated modes (averaged across agents and traffic scenarios). A higher L2 distance indicates greater geometric diversity in generated motions. As shown in the table below, we did not observe a significant drop in mode diversity after alignment.
> >
> > |                               | Reference Model | After Preference Alignment |
> > |-------------------------------|-----------------|-----------------------------|
> > | **Diversity measure: mean pair-wise L2 [m]** |   20.7            | 18.4                        |
> >
> > Nevertheless, we are excited about further investigating mode collapse and exploring mitigation strategies in future work. For example, explicitly modeling and optimizing for multi-modal rankings could help ensure that minority preferences are better captured, maintaining diversity while aligning with human preferences.
> >
> > [1] Xiao, Jiancong, et al. "On the Algorithmic Bias of Aligning Large Language Models with RLHF: Preference Collapse and Matching Regularization." arXiv preprint arXiv:2405.16455 (2024).
> >
> > ***Q5.  How does your OT-distance metric factor in collision / progress / comfort features?***
> >
> > A5. We use the following features [collision status, distance to road boundary, minimum clearance to other road users, control effort, speed] to construct the feature vector $\phi$ when solving the coupling matrix (2).
> >
> > ***Q6.  Do you have qualitative figures that help readers understand why L2 distance drop at the end in Fig 3? Why does L2 distance metric will lead to missed turn in Fig5?***
> >
> > A6.
> > ***L2 distance drop at the end in Fig 3***. We would like to clarify that the L2 distance in Fig. 3 is used as a controlled variable. Specifically, we select sampled traffic simulations from the reference model and analyze the relationship between their group-averaged ADE to the expert demonstrations and their realism. The purpose of this analysis is to demonstrate how L2 distance correlates with model realism.
> >
> > The sharp drop in the blue line of Fig. 3 means that when we try to select sampled traffic simulations with even smaller ADEs, this does not help improve the realism.
> >
> > ***Why does the L2 distance metric lead to missed turn in Fig5?*** When using ADE to rank the generations, we compute the average ADE across all agents in the traffic simulation. In Fig. 5, while the vehicle highlighted near the red circle in the DPA-ADEF figure missed its turn, the overall ADE of the traffic simulation is significantly better than that of the reference model (reference Model ADE: 5.93, DPA-ADEF: 3.17) and slightly better than DPA-OMF (ADE: 3.36).
> >
> > This example demonstrates that optimizing ADE does not necessarily lead to an improvement in realism. ADE primarily captures geometric proximity to expert trajectories but does not account for task-critical aspects.

---

> ### Comment · Reviewer_3Bbi · 2024-11-24
> **response**
>
> Authors have addressed most of my concerns during rebuttal and I am happy to raise my score.

---

> > ### Author Response · Authors · 2024-11-26
> >
> > Dear reviewer 3Bbi,
> >
> > We are glad that our responses could address your questions, and we appreciate your reconsideration on the score of our paper!
> > Thank you again for taking the time to review our paper and providing the insightful comments!
> >
> > Best regards,
> >
> > Authors

---

### Official Review · Reviewer_mv7M · 2024-11-08

**Soundness:** 3
**Presentation:** 3
**Contribution:** 3
**Rating:** 8
**Confidence:** 3

**Summary:**

The paper introduces a method for aligning a token-based motion forecasting model better with demonstrations. The method is based on fine-tuning a pretrained model using a contrastive approach.

**Strengths:**

- **S.1:** The method shows great results on bringing a 1M parameter model up to the performance of larger models.
- **S.2:** The writing is mostly clear.
- **S.3:** The SFT comparison and Fig.4 are interesting.

**Weaknesses:**

- **W.1:** I don't fully understand from the paper how the embedding works, the agent feature encoder. Could you please either give me some implementation details or some better high-level overview?
- **W.2:** Some figures are confusing. Fig.1: What are the orange lines on the left above the motion token pred. model? A-hat is not explained. Fig.3: I don't know how to read this diagram. What's the takeaway? Fig.6: I'm completely lost as to what I'm supposed to do with these.

**Questions:**

- **Q.1:** The writing could use a proofreading pass. There are some minor spelling issues throughout.

---

> ### Author Response · Authors · 2024-11-23
> **Responses to reviewer mv7M [1/2]**
>
> We would like to thank the reviewer for taking the time to review our paper and provide valuable feedback. We are glad that the reviewer thinks our idea is interesting and our approach demonstrates great performance compared to baselines. We are excited to have the chance to address the reviewer’s questions and concerns. These edits will make the paper stronger.
>
> ***Q1.  I don't fully understand from the paper how the embedding works, the agent feature encoder. Could you please either give me some implementation details or some better high-level overview.***
>
> A1. We would like to thank the reviewer for raising this question. Our implementation follows from [1] (specifically Section 3.2.1 and Section 3.2.2). We have added additional details of the scene encoder in Appendix C:
> The scene encoder integrates multiple input modalities, including the road graph, traffic light states, and the trajectory history of surrounding agents. These inputs are first projected into a common latent space through modality-specific encoders. The resulting latent embeddings for each modality are then augmented with learnable positional encodings to preserve spatial and temporal relationships.
> The augmented embeddings are concatenated and passed through a self-attention encoder, which generates a scene embedding for each modeled agent. These scene embeddings are subsequently used by the autoregressive model, via cross-attention, to predict the actions of each agent.

---

> > ### Author Response · Authors · 2024-11-23
> > **Responses to reviewer mv7M [2/2]**
> >
> > ***Q2.  Some figures are confusing. Fig.1: What are the orange lines on the left above the motion token pred. model? A-hat is not explained. Fig.3: I don't know how to read this diagram. What's the takeaway? Fig.6: I'm completely lost as to what I'm supposed to do with these.***
> >
> > A2.  We would like to thank the reviewer for raising these questions and we apologize for the confusion caused by insufficient explanations. We have added the following detailed explanations in the revised paper.
> >
> > **Orange lines in Fig. 1**. In Fig. 1, the orange elements represent components associated with our proposed approach, DPA-OMF. Specifically, the gray dotted lines above the motion token prediction model indicate the reference model’s action distributions at each prediction step. The orange lines illustrate how these probabilities are updated after fine-tuning to align with human preferences.
> >
> > **hat notations**. The hat notations represent the sampled actions during inference. Specifically, during inference, at each prediction step, actions from the previous step are sampled from the predicted distributions. These sampled actions are then used as inputs to the model to predict the conditional probability distribution for the current step's action tokens.
> >
> > **Fig. 3**. Both ADE and our preference score try to measure if a motion generation is close to the expert demonstration. The key difference is that ADE calculates the L2 difference between motion geometries, while our preference score is derived from IRL and measures alignment between occupancy measures. The purpose of Fig. 3 is to show that our preference score correlates more strongly with the realism of generated motions, making it a valid metric for constructing preference rankings.
> >
> > We demonstrate this in a post-selection analysis, where we select sampled traffic simulations from the reference model and analyze the relationship between their group-averaged distance (ADE or preference distance) to the expert demonstration and the realism of these samples (i.e., we control the distance to the expert demonstration, and measure the realism of the selected sampled traffic simulations).
> >
> > In Fig. 3, as ADE decreases, the realism of model generations initially improves. However, beyond a certain point, further reductions in ADE have diminishing returns in terms of realism. In contrast, as we reduce the preference distance, we observe a stronger and more sustained correlation with realism, allowing the preference score to push realism from 0.725 to 0.76.
> >
> > The takeaway is that our preference metric better captures model realism (alignment with expert preferences) than ADE, supporting its use as a basis for constructing preference rankings in our approach.
> >
> > **Left figure of Fig. 6**. The left side of Fig. 6 illustrates how our approach, DPA-OMF, improves the reference model’s performance as the size of the preference dataset increases. A key advantage of DPA-OMF is its ability to leverage expert demonstrations and model generations to automatically construct preference rankings without requiring additional human annotations, making it highly scalable.
> > However, due to limitations in training infrastructure, we were unable to scale the preference data to the desired level to fully maximize performance. Nevertheless, the observed scaling trends demonstrate that the performance of our approach improves with larger preference datasets. This suggests that, with enhanced training resources, our approach could achieve even better results.
> >
> > **Right figure of Fig. 6**. The right side of Fig. 6 illustrates the phenomenon of preference over-optimization, a topic studied in most of the LLM preference alignment works. Since our work shares many connections with LLM alignment, we conducted a similar investigation in the context of multi-agent motion generation. Preference over-optimization tries to understand how much improvements we can gain as we allow the optimized model to deviate further from the reference model. The key takeaways from this plot are twofold: 1) aligning the model with a small preference dataset can actually hurt the performance of the reference model, rather than improving it. 2) if the optimized policy is allowed to deviate from the reference a lot (by applying a smaller weight to the reference model deviation cost during alignment), eventually the performance will degrade, which is consistent with findings in LLMs. However, increasing the amount of preference data can help mitigate this degradation.
> > These two takeaways emphasize the critical role of large-scale preference data in improving imitative token prediction models, and further motivate our approach, which leverages expert demonstrations to construct scalable preference datasets.
> >
> > ***Q3.  The writing could use a proofreading pass. There are some minor spelling issues throughout.***
> >
> > A3.  We would like to thank the reviewer for this comment. We have fixed the typos and grammar mistakes in the revised paper.

---

> ### Comment · Reviewer_mv7M · 2024-11-25
> **These changes make sense to me, updating my score**
>
> I appreciate the authors' changes, and I'm adjusting my score.

---

> > ### Author Response · Authors · 2024-11-26
> >
> > Dear reviewer mv7M,
> >
> > We are glad that our responses could address your questions, and we appreciate your reconsideration on the score of our paper!
> > Thank you again for taking the time to review our paper and providing the insightful comments!
> >
> > Best regards,
> > Authors

---

### Author Response · Authors · 2024-11-23
**General statement [1/2]**

We sincerely thank all the reviewers for their helpful comments and suggestions. We are happy that the reviewers found our paper well-motivated, our idea of preference alignment from occupancy measure matching is novel, and aour approach demonstrates strong results supported by comprehensive ablations. We appreciate this opportunity to address the questions and make improvements to the manuscript. The rebuttal contents are also incorporated in the manuscript (highlighted in blue). Specifically, we made the following major changes:

**G1. Additional experiment to compare the proposed approach with the adversarial preference alignment baseline in which samples from the reference model are treated as negative samples.**

We appreciate the reviewer’s suggestion to compare our approach with prior adversarial preference alignment method, and we have provided both quantitative and qualitative analyses (detailed results and analyses are provided in Appendix F of the updated main manuscript).

Following this suggestion, we compared our method with the AFD approach that treats all samples from the reference model as negative samples. **Our findings indicate that our method outperforms the adversarial preference alignment baseline in terms of the realism of the fine-tuned model, the ability to assign higher likelihood to preferred traffic simulations from the reference model (measured as classification accuracy), and minADE, as shown in Table below**.

| Features         | Classification Accuracy ↑ | Composite Realism ↑ | minADE ↓  |
|------------------|---------------------------|----------------------|-----------|
| **Ours**         | **0.84**                  | **0.739**            | **1.413** |
| **Adversarial AFD**  | 0.52                      | 0.720                | 1.539     |

**Table 3:** The comparison between DPA-OMF with adversarial AFD. Our approach significantly outperforms the adversarial AFD in all metrics.

To further analyze why adversarial preference alignment from demonstrations is less effective, we plotted the negative log-likelihood of expert demonstrations, preferred traffic simulations, and unpreferred traffic simulations in Figure 9 (in Appendix F). The plot shows that the likelihood of expert demonstrations is consistently much higher ($\approx$-205) than that of both preferred and unpreferred samples ($\approx$-245) throughout the alignment process (this stems from the pre-training phase, where expert demonstrations are used to train the reference model), and the likelihoods of preferred and unpreferred samples are very similar. This indicates that the model is unable to capture nuanced differences between preferred and unpreferred samples, leading to suboptimal alignment performance.


**G2. Additional experiments to demonstrate the bias introduced by the heterogeneity of the preference data.**

In G1, we showed that using expert demonstrations as preferred samples and model generations as unpreferred samples results in increasing the likelihood of expert demonstrations without significantly affecting the likelihood of either preferred or unpreferred generated samples.  This suggests that the model struggles to associate the features that make expert demonstrations preferred with the generated preferred samples.

To further explore this, we conducted a separate experiment demonstrating how a discriminative objective using expert demonstrations as positive samples and model generations as negative samples can lead to spurious correlations. Detailed results are included in Appendix G of the updated main manuscript.

In this experiment, we trained a discriminator using a contrastive objective to distinguish between expert demonstrations and model generations. The discriminator achieved a classification accuracy of $0.83$ on the evaluation dataset, indicating it can reasonably classify motions as either expert demonstrations or reference model generations. When the trained discriminator was used to rank pairs of model-generated motions, we observed a pattern: motions with zig-zag trajectories were often classified as unpreferred, while relatively smooth motions were classified as preferred, even when there is un-human like behaviors (e.g., stuck on roads) (see the example in Figure 10 of the updated main manuscript).

This behavior arises because of the heterogeneity of the two data sources: most human demonstrations exhibit smooth motions, while model generations are not constrained by vehicle dynamics. Consequently, the contrastive objective may incentivize the model to pick up this spurious correlation, prioritizing smoothness over other critical attributes.

---

> ### Author Response · Authors · 2024-11-23
> **General statement [2/2]**
>
> **G3. Human study to demonstrate the cost of querying humans for preferences in multi-agent traffic generations.**
>
> To quantify the human cost associated with providing preference rankings for multi-agent traffic simulations, we conducted an Institutional Review Board (IRB)-approved human subject study to measure the effort required (also shown in Appendix H of the updated main manuscript).
>
> In this study, we presented paired traffic simulations to participants and asked them to rank the pairs based on how realistic the simulations were compared to their personal driving experience. We varied the number of traffic agents in the simulations and recorded the time needed to provide rankings.
>
> Five participants ranked 500 pairs of traffic simulations, and the table below summarizes the time required to complete this task. The results show a clear trend: as the number of traffic agents increases, the time required for human annotators to rank simulations grows significantly. Although this study was conducted under time constraints and is not exhaustive, it provides a useful estimate of the human cost for constructing preference rankings at scale. Specifically, for the preference data used in our experiments, the estimated average time required for one human annotator is approximately **633** days.
>
> | Num. of agents in the scene | 1   | 10  | 20  | 40  | 80  |
> |-----------------------------|------|------|------|------|------|
> | Average time used for ranking [s] | 0.7  | 4.9  | 9.8  | 29.4 | 42.1 |
>
> **Table 4**: Average time required for a human to rank traffic simulations.
>
> **This result underscores the practical challenges of scaling preference ranking annotations in multi-agent scenarios, motivating our approach to leverage existing demonstrations to construct preference rankings efficiently.**

---

### Meta-Review · Area_Chair_NUpv · 2024-12-26

**Metareview:**

The authors introduce a novel framework for multi-agent motion generation. Their approach leverages the implicit preference ordering given by expert demonstrations, meaning that they can extract richer supervision from less data. The approach is applied to an autonomous car domain, wherein the authors show this approach scales to 128 agents using a 1 million token prediction model.

The idea is novel, the paper is well explained, and the results appear solid. The reviewers agree the ideas are interesting and that the paper is mostly clear and well-written.

**Additional Comments On Reviewer Discussion:**

The authors took time to respond to reviews, improving some readability issues and adding a comparison with a new baseline suggested by the reviewers. Reviewers increased their score in response to reviewers and believed their concerns were addressed.

---

### Decision · Program_Chairs · 2025-01-22

Accept (Spotlight)